# Learning to Design and Use Tools
# for Robotic Manipulation

**Ziang Liu**\*, **Stephen Tian**\*, **Michelle Guo, C. Karen Liu, Jiajun Wu**
Department of Computer Science
Stanford University, United States
`ziangliu,tians@stanford.edu`

**Abstract:** When limited by their own morphologies, humans and some species of animals have the remarkable ability to use objects from the environment toward accomplishing otherwise impossible tasks. Robots might similarly unlock a range of additional capabilities through tool use. Recent techniques for jointly optimizing morphology and control via deep learning are effective at designing locomotion agents. But while outputting a single morphology makes sense for locomotion, manipulation involves a variety of strategies depending on the task goals at hand. A manipulation agent must be capable of *rapidly prototyping* specialized tools for different goals. Therefore, we propose learning a *designer policy*, rather than a single design. A designer policy is conditioned on task information and outputs a tool design that helps solve the task. A design-conditioned controller policy can then perform manipulation using these tools. In this work, we take a step towards this goal by introducing a reinforcement learning framework for jointly learning these policies. Through simulated manipulation tasks, we show that this framework is more sample efficient than prior methods in multi-goal or multi-variant settings, can perform zero-shot interpolation or fine-tuning to tackle previously unseen goals, and allows tradeoffs between the complexity of design and control policies under practical constraints. Finally, we deploy our learned policies onto a real robot. Please see our supplementary video and website at `https://robotic-tool-design.github.io/` for visualizations.

**Keywords:** tool use, manipulation, design

## 1 Introduction

Humans and some species of animals are able to make use of tools to solve manipulation tasks when they are constrained by their own morphologies. Chimpanzees have been observed using tools to access food and hold water [1], and cockatoos are able to create stick-like tools by cutting shapes from wood [2] with their beaks. To flexibly and resourcefully accomplish a range of tasks comparable to humans, embodied agents should also be able to leverage tools.

But critically, while any object in a human or robot's environment is a potential tool, not every object is directly a useful aid for the task goal at hand. How can a robotic system also acquire the extraordinary ability of animals to create an appropriate tool to help solve a task after reasoning about a scene's physics and its own goals? In this work, we investigate not only how agents can perform control using tools, but also how they can learn to *design* appropriate tools when presented with a particular task goal, such as a target position or object location.

Prior works have studied joint learning of agent morphologies and control policies for locomotion tasks [3, 4, 5, 6, 7]. However, these approaches optimize designs for a single, predefined *generic* goal, such as maintaining balance or forward speed. In this work, we take a step towards agents that can learn to rapidly prototype *specialized* tools for different goals or initial configurations.

---

\*indicates equal contribution.

7th Conference on Robot Learning (CoRL 2023), Atlanta, USA.

We propose tackling this challenge by learning *designer* and *controller* policies, that are conditioned on task information, solely from task rewards via reinforcement learning (RL). We find that when trained with a multi-stage Markov decision process (MDP) formulation, these policies can be efficiently learned together in a high-dimensional combined space. We train agents on multiple instantiations of each task so that they can learn to produce and manipulate designs best suited to each situation.

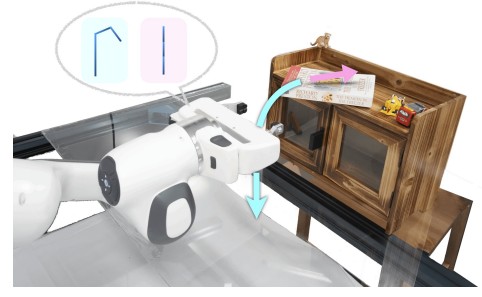

Figure 1: A robot may need to use different tools to fetch an out-of-reach book (blue) or push it into the bookshelf (pink). It should *rapidly prototype* the tool it needs.

We argue that there are three important properties of an embodied agent that designs and uses tools in realistic settings. Firstly, it must be able to learn design and control policies without explicit guidance, using signals specified only on task progress. Furthermore, it should form specialized tools based on the task goal at hand, as motivated by Figure 1. Finally, it should adjust to real-world constraints, rather than creating infeasible designs.

Our main contribution is a learning framework for embodied agents to design and use tools *based on the task at hand*. We demonstrate that our approach can jointly learn these policies in a sample-efficient manner from only downstream task rewards for a variety of simulated manipulation tasks, outperforming existing stochastic optimization approaches. We empirically analyze the generalization and few-shot finetuning capabilities of the learned policies. By introducing a tradeoff parameter between the complexity of design and control components, our approach can adapt to fit constraints such as material availability or energy costs. Finally, we demonstrate the real-world effectiveness of the learned policies by deploying them on a real Franka Panda robot.

## 2   Related Work

**Computational approaches to agent and tool design.** Many works have studied the problem of optimizing the design of robotic agents, end-effectors, and tools via model-based optimization [8, 9], generative modeling [10, 11], evolutionary strategies [3, 5], stochastic optimization [12], or reinforcement learning [13]. Li et al. [14] use differentiable simulation to find parameters of a tool that are robust to task variations. These methods provide feedback to the design procedure by executing pre-defined trajectories or policies, or performing motion planning. In contrast, we aim to *jointly* learn control policies along with designing tool structures.

In settings where the desired design is known but must be assembled from subcomponents, geometry [15] and reinforcement learning [16] have been used to compose objects into tools. In this work, we address the fabrication stage of the pipeline using rapid prototyping tools (e.g. 3D printing).

**Learning robotic tool use.** Several approaches have been proposed for empowering robots to learn to use tools. Learning affordances of objects, or how they can be used, is one common paradigm [17, 18, 19, 20]. Noguchi et al. [21] integrate tool and gripper action spaces in a Transporter-style framework. Learned or simulated dynamics models [22, 23, 24, 25] have also been used for model-based optimization of tool-aided control. These methods assume that a helpful tool is already present in the scene, whereas we focus on optimizing tool design in conjunction with learning manipulation, which is a more likely scenario for a generalist robot operating for example in a household.

**Joint optimization of morphology and control.** One approach for jointly solving tool design and manipulation problems is formulating and solving nonlinear programs, which have been shown to be especially effective at longer horizon sequential manipulation tasks [26, 27]. In this work, we aim to apply our framework to arbitrary environments, and so we select a purely learning-based approach at the cost of increasing the complexity of the search space.

Reinforcement learning and Bayesian optimization (BO) [28] based approaches have also been applied to jointly learn morphology and control. These include policy gradient methods with either separate [29] or weight-sharing [30] design and control policies or methods that consider the agent

design as a computational graph [31]. Evolutionary or BO algorithms have been combined with control policies learned via RL [32] to solve locomotion and manipulation tasks. Luck et al. [4] use an actor-critic RL formulation with a graph neural network (GNN) value function for improved sample efficiency. Pathak et al. [33] learn to design modular agents by adding morphology-modifying actions to an MDP. Yuan et al. [7] provide a generalized formulation with a multi-stage MDP and GNN policy and value networks. These methods have demonstrated promising performance on locomotion tasks. In this work, we focus on developing tools for manipulation, where the challenge is learning designer and controller policies that can create and operate different tools *depending on the task variation at hand*, and it is less feasible to use actuated joints as design components.

## 3 Problem Setting

Our framework tackles learning tool design and use for agents to solve manipulation problems, without any supervision except for task progress. We represent the agent's environment as a two-phase MDP consisting of the "design phase" and "control phase". We use environment interactions from both phases to jointly train a designer policy and controller policy.

At the start of each episode, the environment begins in the **design phase**, visualized at the top of Figure 2. During the design phase, the action $a_d \in \mathcal{A}_D$ specifies the parameters of the tool that will be used for the rest of the episode. The environment state $s_0 \in \mathcal{S}_{task}$ consists of a vector of task observations: the positions and velocities of objects in the scene and the Cartesian end-effector position of the robot if present.

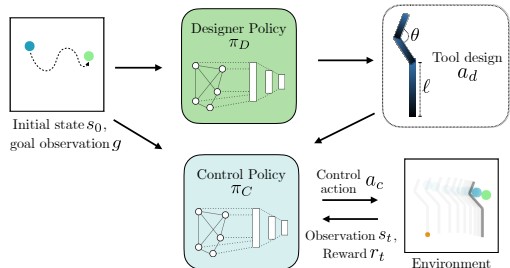

After a single transition, the MDP switches to the **control phase**, illustrated in the lower half of Figure 2. During the control phase, the actions $a_c \in \mathcal{A}_C$ represent inputs to a position or velocity controller actuating the base of the previously designed tool. The control phase state

Figure 2: Solving a task using learned designer and controller policies. During the design phase, the designer policy observes the task at hand and outputs the parameters for a tool. In the control phase, the controller policy outputs motor commands given the tool structure, task specification, and observation.

is the concatenation of the task observation $s_t \in \mathcal{S}_{task}$ and the previously taken design action $a_d$. That is, the control state space $\mathcal{S}_C = \mathcal{S}_{task} \times \mathcal{A}_D$. The agent receives rewards $r_t$ at each timestep $t$ based on task progress (e.g., the distance of an object being manipulated to the target position). The control phase continues until the task is solved or a time limit is reached, and the episode then ends.

To learn design and control policies for multiple goals, we condition our policies on a supplied parameter $g$ from a goal or task variation space $\mathcal{G}$. In this paper, we choose goals that represent the final position of an object to be manipulated or the number of objects that must be transported. The objective is then to find the optimal *goal-conditioned* designer and controller policies $\pi_D^\star(a|s,g)$, $\pi_C^\star(a|s,g)$ that maximize the expected discounted return of a goal-dependent reward function $R(s,a,g)$.

## 4 Instantiating Our Framework

Next, we concretely implement our framework toward solving a series of manipulation tasks. Specifically, we select a **tool design space**, **policy learning procedure**, and **auxiliary reward function**.

**Tool design space.** The design space significantly impacts the difficulty of the joint design and control optimization problem. When the set of possible designs is large but many of them are unhelpful for *any* task, the reward signal for optimization is sparse. Thus, we would like to select a design parameterization that is low-dimensional but can also enable many manipulation tasks. Furthermore, we prefer designs that are easy to deploy in the real world.

Toward these goals, we consider tools composed of rigid links. While simple, we find that this parameter space includes tools that are sufficient to help solve a variety of manipulation tasks. They

can also be easily deployed in the real world on soft robots [12] or through rapid fabrication techniques like 3D printing. However, we note that our framework is not limited to this design space.

For example, the parameterization used in our 2D environments consists of three links attached end-to-end as shown in Figure 2, where a tool is represented by a vector $[l_1, l_2, l_3, \theta_1, \theta_2] \in \mathbb{R}^5 = \mathcal{A}_D$, where $l$ represents each link length and $\theta$ represents the relative angle between the links.

**Policy learning.** Similarly to Yuan et al. [7], we interactively collect experience in the environment using the design and control policies, where each trajectory spans the design and control phase. We then train the policies jointly using proximal policy optimization (PPO) [34], a popular policy gradient method. We adopt the graph neural network (GNN) policy and value function architecture from Yuan et al. [7]. When training in goal-conditioned environments, we supply the policies with randomized goals sampled from the environment for each interaction episode.

**Auxiliary reward.** An embodied agent that creates and uses tools in the real world must also consider resource constraints. Many prior co-optimization procedures assume that actuated joints or body links can be arbitrarily added to the agent morphology. However, as an example, when an agent solves manipulation tasks in a household environment, it may not have access to additional motors or building materials. On the other hand, when possible, constructing a larger tool may reduce the amount of power expended for motor control, especially if a task must be completed many times.

We enable our framework to accommodate preferences in this trade-off between design material cost and control energy consumption, proxied by end-effector velocity, using a parameter $\alpha$ that adjusts an auxiliary reward that is added to the task reward at each environment step:

$$r_{\text{tradeoff}} = K \left[ 1 - \left( \frac{\alpha \cdot d_{\text{used}}}{d_{\text{max}}} + \frac{(1 - \alpha) \cdot c_{\text{used}}}{c_{\text{max}}} \right) \right], \tag{1}$$

where $K$ is a scaling hyperparameter, $\alpha \in [0, 1]$ controls the emphasis on either the control or design component, $d_{\text{used}}$ and $d_{\text{max}}$ represent the utilized and maximum possible combined length of the tool components in the design respectively, and $c_{\text{used}}$ and $c_{\text{max}}$ represent the control velocity at the current step and the maximum single-step control velocity allowed by the environment. Intuitively the agent favors using less material for tool construction when $\alpha$ is large, and less energy for the control policy when $\alpha$ is small. Except in Section 5.3, we use $K = 0$ to isolate this reward's effects.

# 5 Experiments

For our experiments, we introduce three 2D manipulation environments in the Box2D simulator [35] and three 3D environments in PyBullet [36]. These tasks showcase the advantages of using different tools when there does not exist a single tool that can solve all instances. The six tasks are shown in Figure 3. For each task, we initialize the designed tool by matching a fixed point on the tool to a fixed starting position regardless of the goal. During the control phase, we simulate a scenario in which a robot has grasped the tool and manipulates it, via end-effector velocity control in 2D or position control in

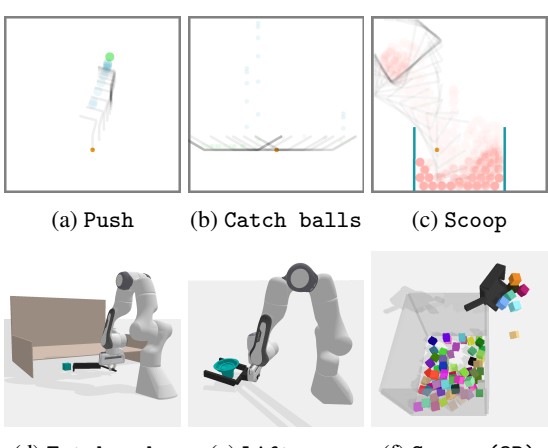

(a) `Push`   (b) `Catch balls`   (c) `Scoop`

(d) `Fetch cube`   (e) `Lift cup`   (f) `Scoop (3D)`

Figure 3: Simulated manipulation environments.

3D. All 2D tasks use the 3-link chain parameterization described in Section 4. A short description of each task is as follows, with additional details in the Appendix:

- `Push` (2D): Push a round puck using the tool such that it stops at the specified 2D goal location.
- `Catch balls` (2D): Use the tool to catch three balls that fall from the sky. The agent's goal is to catch all three balls, which start from random locations on the $x$-$y$ plane.
- `Scoop` (2D): Use the tool to scoop balls out of a reservoir containing 40 total balls. Here we specify goals of scooping $n \in \{1, 2, ..., 7\}$ balls.

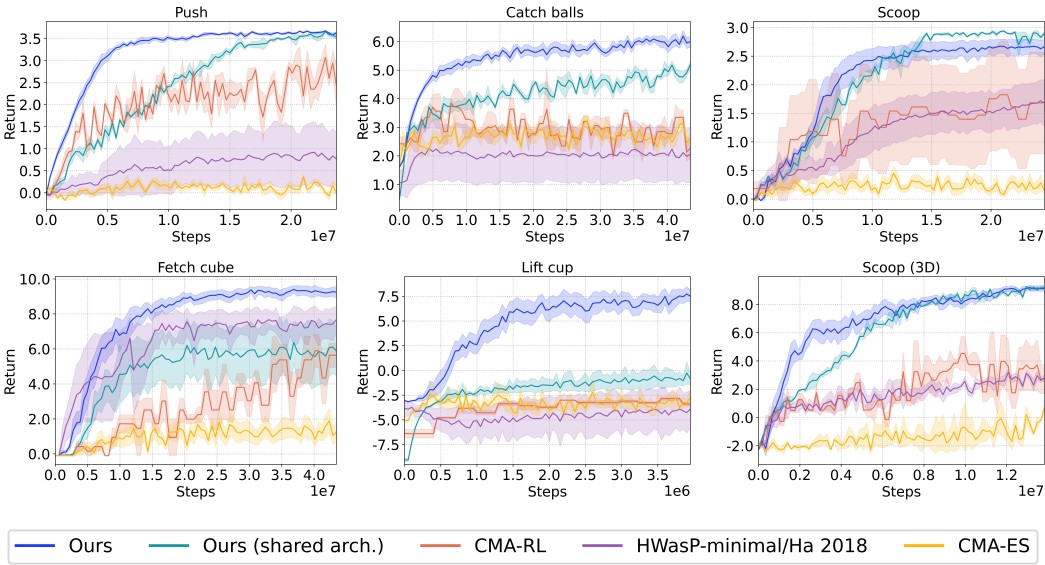

Figure 4: Learning curves for our framework, prior methods, and baselines. Across all tasks, our framework achieves improved performance and sample efficiency. Shaded areas indicate standard error across 6 random seeds on all methods, except the `scoop (3D)` task where we use 3 seeds due to computational constraints.

- `Fetch cube (3D)`: Use the tool to retrieve an object randomly positioned beneath an overhang. This task is challenging because the end effector is restricted to a $0.8\text{m} \times 0.2\text{m}$ region of the $x$-$y$ plane to avoid collision with the overhang. The tool is a three-link chain where each link is a box parameterized by its width, length, height, and relative angle to its parent link.
- `Lift cup (3D)`: Use the tool to lift a cup of randomized geometry from rest to a certain height. This task requires careful tool design to match cup geometry. The tool is a four-link fork with two prongs symmetrically parameterized by their separation, tilt angle, width, length, and height.
- `Scoop (3D)`: An analog of the 2D `scoop` task (with the same goal space), but the tool in 3D is a six-link scoop composed of a rectangular base plate parameterized by its length and width, and four rectangular side plates attached to each side of the base plate, parameterized by their height and relative angle to the bottom plate. A fixed-dimension handle is attached to one side plate.

In our experiments, we analyze whether the instantiation of our framework on these manipulation tasks has the following four desirable properties:

- Can our framework jointly learn designer and controller policies in a sample-efficient manner, using just rewards based on task progress?
- Do learned designer and controller policies generalize or enable fine-tuning for unseen goals?
- Can the adjustable parameter $\alpha$ enable agents to trade off design and control complexity?
- Can designer and controller policies learned by our framework be deployed on a real robot?

## 5.1 Evaluating sample efficiency

Sample efficiency is critical for a performant joint tool design and control learning pipeline, as many sampled designs will be unhelpful for *any* goal or initialization. As prior joint optimization works do not handle situations where diverse designs may be produced depending on the particular task variation, we compare to the following prior methods and baselines (details in Appendix B.2):

- **Bi-level optimization (CMA-RL):** This follows the common bi-level paradigm for joint optimization [4]. We use CMA-ES [37] to perform outer-loop stochastic design optimization and learn design-conditioned policies with PPO in the inner loop, as the reward signal for design.
- **Hardware-as-Policy Minimal [31]/Ha [30]:** The variant of HWasP without differentiable physics assumptions (HWasP-minimal) and the method from Ha [30] both perform policy-gradient optimization of a single set of design parameters together with the control policy jointly using RL.

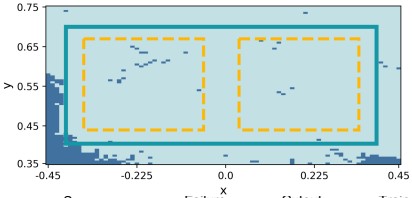 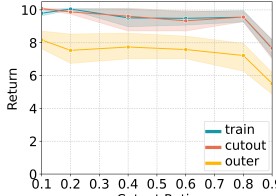 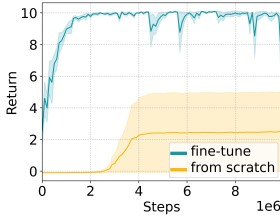

(a) Initialization ranges and zero-shot performance when cutting out 60% of the area of the entire possible training region.

(b) Returns for policies trained with varying relative cutout region area.

(c) Fine-tuning performance compared to learning from scratch across 4 target goals.

Figure 6: We find that our policies can solve instances of the `fetch cube` task unseen at training time either zero-shot or by rapid finetuning. In (a), we plot the goal regions along with zero-shot policy performance. Areas within the dotted yellow borders and outside the teal region are unseen during training. The region within the teal border (but outside cutout regions) is the training region. Training curves are averaged over 3 seeds; shaded regions show standard error.

- **Single-trajectory CMA-ES baseline:** We optimize a set of design and control actions for an episode independent of the goal or starting task state, demonstrating the required policy reactivity.
- **Ours (shared arch.):** An ablation of our framework that uses a single policy network with design and control heads, demonstrating the importance of separate designer and controller architectures.

In Figure 4, we compare the learning curves of our method and demonstrate results for all six tasks. Our method strongly outperforms the prior methods and baselines, achieving superior final performance in fewer samples. It does so by producing specialized tools to solve each task instance, while the other methods optimize for a single design across all task variations. The ablation shows the importance of learning separate designer and controller policies.

We present qualitative examples of tool designs outputted for different goals on the `push` task in Figure 5. The designer policy outputs a range of tools depending on the goal location.

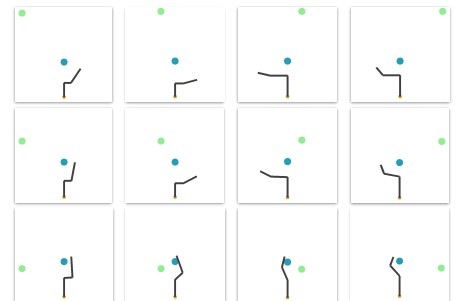

Figure 5: Tool designs outputted by a single learned designer policy for the `push` task as the goal position, in green, varies. The range of outputted designs enable the agent to push the ball in the desired direction.

## 5.2 Generalization to unseen task variations

In simulation, agents can experience millions of trials for a range of task variations. However, when deployed to the real world, designer and controller policies cannot be pre-trained on all possible future manipulation scenarios. In this section, we test the ability of our policies to generalize to task variations unseen during training. For these experiments, we focus on the `fetch cube` task because it has an initial pose space that can be manipulated in a semantically meaningful way. We train policies using our framework on a subset of initial poses from the entire initial pose space by removing a region of the space, which we call the "cutout" region. (see Figure 6a). Then, we evaluate the generalization performance of learned policies on the initial poses from the cutout region and outside the training region in two scenarios.

**Zero-shot performance.** In the first scenario, we test the ability of the designer and controller policy to tackle a previously unseen initial pose directly. Using a policy trained on the initial pose space with a cutout region removed, we evaluate the zero-shot performance on unseen initial poses.

In Figure 6a, we visualize the zero-shot performance of our design and control policies on initial poses across the entire environment plane, finding that our policies are able to solve even task variations outside the training region boundaries. We also analyze how decreasing the number of possible training poses affects generalization performance. In Figure 6b, we show the returns over the training region, cutout region, and the regions outside of the training region, as the size of the cutout region for training poses varies.

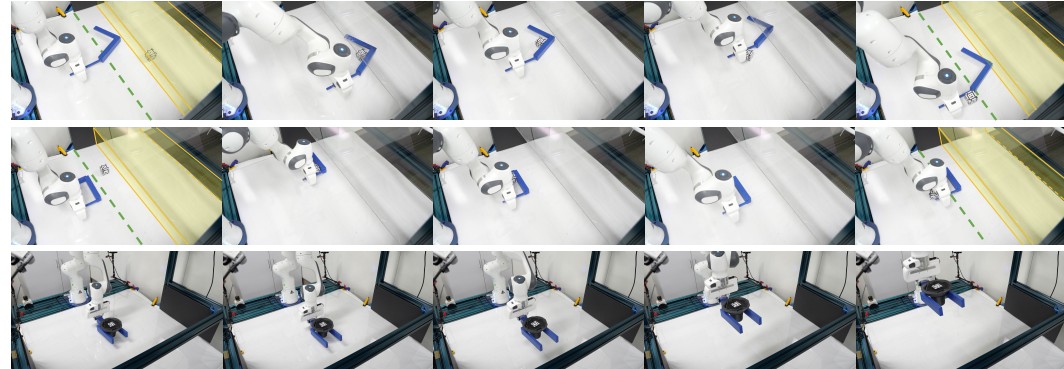

Figure 8: Real world rollouts for the `fetch cube` (top) and `lift cup` (bottom) tasks. For `fetch cube`, we show the task success threshold in green and the volume covered by the overhang in yellow. The design policy generates specialized tools, and the control policy adjusts its strategy to use the tool to complete each task.

When the cutout region is very small, the performance of our learned policies on seen and unseen poses is similar. As the area of the cutout region increases, the performance on unseen goals degrades gracefully and can still solve a significant portion of unseen tasks.

**Fine-tuning performance.** Sometimes, new task variations cannot be achieved zero-shot by designer and controller policies. In this section, we test whether our design policies and controller policies can still serve as good *instantiations* for achieving these variations. We test this by pre-training policies with our framework on the entire training region and fine-tuning them to solve initial poses outside that region.

In Figure 6c, we show the results of the fine-tuning experiment. We find that even for poses that are far away from the initial training region, our policies are able to learn to solve the task within a handful of gradient steps, and is much more effective than learning from scratch.

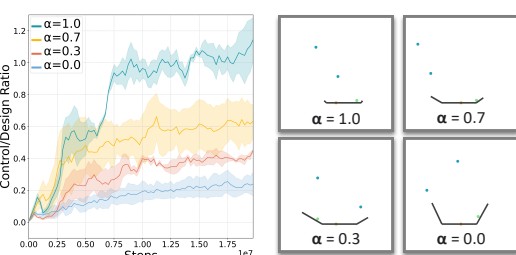

(a) Control/Design ratio with different $\alpha$.   (b) Tools produced at different $\alpha$ values.

Figure 7: Examples of tools generated by varying the tradeoff parameter $\alpha$. As $\alpha$ increases, the created tools have shorter links at the sides to decrease material usage. With lower $\alpha$, large tools reduce the control policy's required movement.

### 5.3 Trading off design and control complexity

In this section, we aim to determine whether our introduced tradeoff parameter $\alpha$ (defined in Equation 1) can actualize preferences in the tradeoff between design material cost and control energy consumption. For this experiment, we focus on the `catch balls` task, because the tradeoff has an intuitive interpretation in this setting: a larger tool can allow the agent to catch objects with minimal movement, while a smaller tool can reduce material cost but requires a longer trajectory with additional energy costs. We train four agents on `catch balls` with different values of $\alpha$.

In Figure 7a, we plot the ratio $\frac{d_{used}/d_{max}}{c_{used}/c_{max}}$, where $d$ represents the combined length of all tool links and $c$ is the per-step control velocity. $d_{max}$ and $c_{max}$ indicate the maximum tool size and control velocity allowed by the environment. We find that this ratio indeed correlates with $\alpha$, which indicates that agents that are directed to prefer saving either material or energy are doing so, at the cost of the other. We also visualize the outputted tool designs in Figure 7b.

### 5.4 Evaluation on a real robot

Next, we provide a demonstration of our pipeline on a real robot by transferring learned policies in two of the 3D environments, `fetch cube` and `lift cup`, directly to the real world. We use a Franka Panda 7-DoF robot arm equipped with its standard parallel jaw gripper (Figure 11). Five RealSense D435 RGBD cameras perform object tracking using AprilTags [38].

**Fabricating tools.** In order to evaluate our policies in the real world, we fabricate the tools that are outputted by the design policy via 3D printing. Specifically, based on an initial state observation and/or goal, we convert tool parameters from design policy outputs into meshes and construct them from PLA using consumer Ender 3 and Ender 5 fused deposition modeling (FDM) 3D printers. See Figure 9 for examples of printed tools generated by our policy.

**Real robot evaluation.** For the `fetch cube` and `lift cup` tasks, we evaluate policies on four initial cube positions and four cup geometries respectively. For each task instance, we fabricate the designed tool and evaluate control over five trials. We report success rates for each tool in Table 1, finding that our policies successfully transfer their performance to the real world.

Figure 8 shows qualitative examples of real rollouts. In the top row, the robot designs an 'L'-shaped hook. Because the end-effector has a constrained workspace due to the risk of collision with the overhang, the robot employs a two-stage strategy – first hooking the block from under the overhang, and then using the backside of the tool to drag it closer to itself. In the second row, the robot uses a 'U'-shaped tool to quickly retrieve the nearby block and finally pushes it to the goal with the gripper fingers when it is within reach. This exemplifies that our framework flexibly allows an agent to use its original morphology in combination with designed tools when needed. Finally, in the `lift cup` task, the design policy selects appropriate distances between the tool arms and an angle of approach to hold the cup securely when elevated.

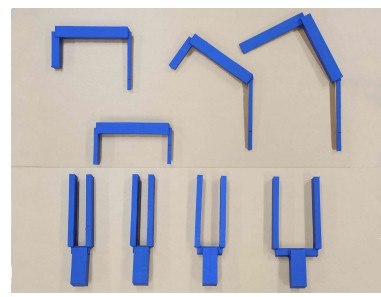

Figure 9: 3D-printed tools for `fetch cube` (top) and `lift cup` (bottom).

For `fetch cube`, we further analyze the performance of the control policy when using tools for initial positions that they were **not directly designed for**. Specifically, we take the four generated tools and evaluate how well the control policy can use them to solve 12 tasks on a 12cm × 85.6cm grid of initial cube positions.

| Tool number | 1 | 2 | 3 | 4 |
|---|---|---|---|---|
| `Fetch cube` (single init.) | 5/5 | 5/5 | 5/5 | 5/5 |
| `Fetch cube` (grid inits.) | 10/12 | 4/12 | 6/12 | 5/12 |
| `Lift cup` | 5/5 | 5/5 | 5/5 | 5/5 |

Table 1: **Real world evaluation performance.** "Single init" denotes the evaluation of a tool on the initial environment state it was designed for. "Grid of inits" evaluates the same tool on a range of initializations.

The results are presented in Table 1. While the control policy can reuse tools to solve new tasks, no tool can solve all the tasks. Tool 1 solves the most tasks, but tool 4 solves both tasks that tool 1 cannot. We also find that the policy takes a greater number of timesteps to solve each task with tool 1 compared to tools specialized for those initializations. These experiments demonstrate that a diverse set of tools is indeed important for different variations of this task.

## 6  Conclusion

We have introduced a framework for agents to jointly learn design and control policies, as a step towards generalist embodied manipulation agents that are unconstrained by their own morphologies. Because the best type of tool and control strategy can vary depending on the goal, we propose to learn designer and controller policies to generate useful tools based on the task at hand and then perform manipulation with them. Our work is a step towards building embodied agents that can reason about novel tasks and settings and then equip themselves with the required tools to solve them, without any human supervision. Such systems may lead the way towards autonomous robots that can perform continuous learning and exploration in real-world settings.

**Limitations.** In this work, we focus on rigid, non-articulated tools composed of linked primitive shapes. A promising direction for future work is to explore other parameterizations. In addition, we do not address fabrication: we use 3D-printing for prototyping and do not consider the problem of constructing tools from a set of available objects. Finally, we focus on tool geometry, but consideration of tool stability and applied forces could lead to improved real-world performance.

**Acknowledgments**

We thank Josiah Wong as well as anonymous reviewers for helpful feedback on earlier versions of the manuscript, Samuel Clarke for recording the video voiceover, and the Stanford Product Realization Lab for 3D printing resources and advice. The work is in part supported by ONR MURI N00014-22-1-2740, the Stanford Institute for Human-Centered AI (HAI), Amazon, Autodesk, and JPMC. ST and MG are supported by NSF GRFP Grant No. DGE-1656518.

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

# A  Environment Details

Here we provide additional details for our simulation environments. An unabridged version of the description from Section 5 is as follows:

- `Push (2D)`: Push a round puck using the tool such that it stops at the specified goal location. The goal space is a subset of 2D final puck locations $\mathcal{G} \subset \mathbb{R}^2$, and the control action space $\mathcal{A}_C \in \mathbb{R}^2$ specifies the $x$ and $y$ tool velocities.

- `Catch balls (2D)`: Use the tool to catch three balls that fall from the sky. The agent's goal is to catch all three balls, which start from varying locations on the $x$-$y$ plane. We use a 1-dimensional control action space that specifies the $x$ velocity of the tool at each step.

- `Scoop (2D)`: Use the tool to scoop balls out of a reservoir containing 40 total balls. Here we specify goals of scooping $x \in \{1, 2, ..., 7\}$ balls. The control action space $\mathcal{A}_C \in \mathbb{R}^3$ specifies the velocity of the rigid tool in $x$ and $y$ directions, along with its angular velocity.

- `Fetch cube (3D)`: Use the tool to retrieve an object randomly positioned beneath a vertical overhang. This task is additionally challenging because the position of the robot end effector is restricted to a rectangular region in the $x$-$y$ plane of dimensions 0.8m × 0.2m to avoid collision with the overhang. The tool is a three-link chain where each link is a box parameterized by its width, length, and height. The design space also includes the relative angle between two connected links, with a total of $n = 11$ parameters. The control action space $\mathcal{A}_C \in \mathbb{R}^3$ represents a change in end-effector position.

- `Lift cup (3D)`: Use the tool to lift a cup of randomized geometry from the ground into the air. This task requires careful design of the tool to match cup geometry. The tool is a four-link fork with two prongs parameterized by the separation, tilt angle, width, length, and height of the prongs. The same parameters are applied to both prongs to maintain symmetry. The handle dimensions are fixed. The design space has $n = 5$ parameters. $\mathcal{A}_C \in \mathbb{R}^3$ represents a change in end-effector position.

- `Scoop (3D)`: A 3D analog of the 2D `scoop` task. This task has the same goal space as the 2D `scoop` task, but the tool in 3D is a six-link scoop composed of a rectangular bottom plate parameterized by its width and length, and four rectangular side plates attached to each side of the bottom plate. Each side plate is parameterized by its height and relative angle to the bottom plate. A handle with fixed dimensions is attached to one of the side plates. There are $n = 10$ total design parameters. $\mathcal{A}_C \in \mathbb{R}^6$ represents a change in end-effector pose.

Our task selection is motivated by real-world tasks that a robot may need to perform for example in a home robot setting. Examples include:

- Fetching objects (fetch cube): retrieving fallen or misplaced objects e.g. underneath sofas, tables, or chairs in homes, or tight spaces like inside cars, enabling scene "resets" from states where objects are lost for continuous robotic learning settings

- Scooping (scoop 2D, 3D): manipulating granular materials such as rice, beans, cereals or measuring and transferring liquids for cooking

- Pushing (push): home robotics applications when robots are impeded by obstacles such as countertops, tables, beds; industrial applications to aligning and grouping objects for robotic packing

- Lifting objects (lift cup): Transporting objects that are too large for a parallel jaw gripper or for a suction gripper to stably grasp, for example, pots and pans, garbage cans, kitchen appliances; or risky to grasp directly, for example, in high temperature industrial welding and forging

# B Experimental Details

## B.1 Training Hyperparameters & Architecture Details for Our Framework

In Tables 5, 6, 7, 8, 9, and 10, we provide detailed hyperparameters for our framework for each environment. Unless otherwise specified, we use the neural network architectures for the design policy, control policy, and value function from [7].

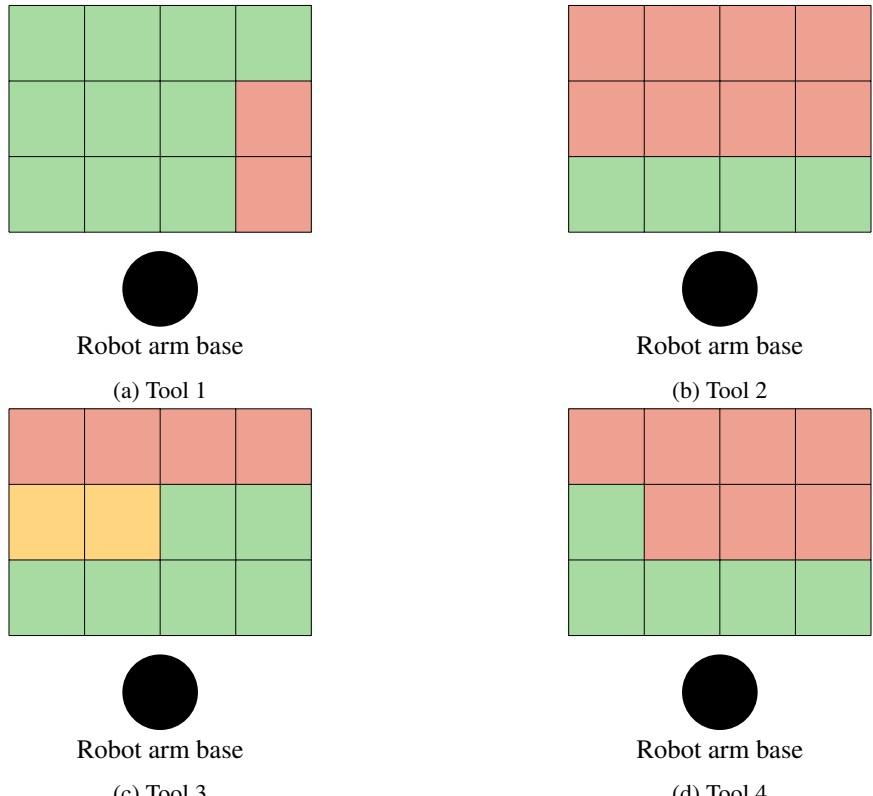

(a) Tool 1

(b) Tool 2

(c) Tool 3

(d) Tool 4

Figure 10: Heatmaps of success and failures for real world `fetch cube` trials where fixed tools are used for a range of initializations. Recall that we test the policies on a $3 \times 4$ grid of initial positions that span a total dimension of $12\text{cm} \times 85.6\text{cm}$, for a total of 12 trials per tool. This means that evaluations were performed every 6cm along one dimension and 28.53cm along the other. The grid here is a top-down view of the robot workspace and directly maps to the set of 2D initial cube locations tested for each tool, where the base of the robot is at the bottom of each diagram. Here green indicates a success, red indicates failure, and orange indicates a failure where the final cube position is within 5cm of success.

## B.2 Training Hyperparameters & Architecture Details for Baselines

For the CMA-ES baseline, we perform hyperparameter sweeps for a fair comparison with our framework. For the CMA-RL baseline, we use the same set of best performing hyperparameters for the outer CMA-ES loop. The tested hyperparameter configurations for each baseline are listed in Table 2. Except model architecture differences, we use the same optimization hyperparameters for Ours, Ours(shared arch.), and HWasP-minimal. Note that we control for the number of network parameters in the "shared arch" ablation – notably, we used MLP policies implemented in Stable Baselines 3 [39] and ensure that the number of trainable network parameters is either the same or strictly larger than in our method across all tasks.

## B.3 Computational Resources

We train each of our policies using a single GPU (NVIDIA RTX 2080Ti or TITAN RTX) and 32 CPU cores. The total wall clock training time varies per environment from 2

| Method | Hyperparameters | Values |
|---|---|---|
| CMA-ES | Population Size | 10, **24**, 100, 1000 |
| | Initial Stdev | **0.1**, 1.0, 10.0 |
| | Center Learning Rate | 0.01, 0.1, **1.0** |
| | Covariance Learning Rate | 0.01, 0.1, **1.0** |
| | Rank $\mu$ Learning Rate | 0.01, 0.1, **1.0** |
| | Rank One Learning Rate | 0.01, 0.1, **1.0** |
| CMA-RL | Poicy Net | (256, 256, 256, 256, 256) |
| | Value Net | (256, 256, 256, 256, 256) |
| | Learning Rate | 3e-4 |
| | Batch Size | (50000, 20000(3D scoop)) |
| | Minibatch Size | 2000 |
| Ours(shared arch.) | Poicy Net | (256, 256, 256, 256, 256) |
| | Value Net | (256, 256, 256, 256, 256) |
| | Learning Rate | 1e-4 |
| | Batch Size | (50000, 20000(3D scoop)) |
| | Minibatch Size | 2000 |

Table 2: We tune over these values for hyperparameters of baseline methods. Bolded values indicate the best performing settings for CMA-ES, which we use in our comparisons.

hours for the Catch Balls environment to 24 hours for the Scoop(3D) environment. We detail the training/inference time for our model and baselines on the fetch cube task:

| | Ours | Ours (shared arch) | CMA-RL | HWasP-minimal/Ha 2018 | CMA-ES |
|---|---|---|---|---|---|
| Training time | 2.12e+2 | 7.29e+2 | 1.18e+3 | 2.98e+2 | 3.46e+2 |
| Inference time | 1.68e-3 | 5.24e-4 | 5.32e-4 | 1.75e-3 | N/A |

The training time represents the wall clock time in minutes needed to train the model for $10^7$ environment steps. The inference time denotes the wall clock time in seconds needed to perform one forward pass through the model. These results are generated using a NVIDIA Titan RTX GPU and an Intel Xeon Gold 5220 CPU.

Figure 11: Real world setup. We use a Franka Panda arm and five RealSense D435 cameras for tracking. The cube pictured, used for the *fetch cube* task has a side length of 5cm.

## B.4 Generalization
## to Unseen Goals Experiment Details

For `Fetch cube`, the rectangular region of initial poses is defined by $x \in [-0.395, 0.395]$ and $y \in [0.4, 0.7]$. The cutout region corresponds to two disconnected rectangular patches contained in the training region defined by $x_1 \in [-0.350, -0.045]$, $y_1 \in [0.434, 0.666]$ and $x_2 \in [0.045, 0.350]$, $y_2 \in [0.434, 0.666]$ respectively.

**Zero-shot performance.** We train six policies using our framework where the cutout region removes a fraction of the total training area equal to 0.1, 0.2, 0.4, 0.6, 0.8, and 0.9 respectively.

**Fine-tuning performance.** For the fine-tuning experiment, we specifically select four initializations that we find our policies do not complete successfully zero-shot: $\{(-0.167, 0.367), (-0.129, 0.357), (0.430, 0.493), (0.415, 0.610)\}$.

## B.5 Trading off Design and Control Complexity Experiment Details

We train four agents independently on `catch balls`, setting the value of $\alpha$, the tradeoff reward parameter defined in Equation 1, to 0, 0.3, 0.7, and 1.0 respectively.

## B.6  Real Robot Experiment Details

For our real-world experiments, we use a Franka Emika Panda arm. We control the robot using an impedance controller from the Polymetis [40] library.

Tools are 3D printed using polylactic acid (PLA) on commercially available Ender 3 and Ender 5 printers with nozzle diameter 0.4mm. We print using a layer height of 0.3mm and 10% infill. We perform slicing using the Ultimaker CURA software.

We roll out each policy for 100 environment steps or until a success is detected.

For the `fetch cube` task, we measure the success based on whether the center of mass of the 5cm cube is closer than 0.5m from the base of the robot. Please see Table 3 for per-tool details. The tool images are shown in Figure 9, from left to right: Tools 4, 2, 3, 1 respectively.

| Tool | Initial cube position (x, y, z) |
|------|-------------------------------|
| Tool 1 | (-0.110, -0.803, 0.025) |
| Tool 2 | (-0.339, -0.588, 0.025) |
| Tool 3 | (0.155, -0.731, 0.025) |
| Tool 4 | (0.211, -0.633, 0.025) |

Table 3: Initial cube positions corresponding to tools fabricated in real experiments.

For the `lift cup` task, we measure success based on whether the cup has been lifted higher than 0.4m off of the plane of the workspace. Please see Table 4 for per-tool details. The tool images are shown in Figure 9, from left to right: Tools 1, 2, 3, 4.

| Tool | Cup geometry parameters (length/width, height) |
|------|-----------------------------------------------|
| Tool 1 | (0.3, 0.6) |
| Tool 2 | (0.3, 0.9) |
| Tool 3 | (0.5, 0.8) |
| Tool 4 | (0.9, 0.6) |

Table 4: Cup geometry parameters corresponding to tools fabricated in real experiments. Note that the length and width parameters share a single value.

We also present detailed results for the `fetch cube` experiments using tools generated for a specific initial position for a range of initializations. Recall that we test the policies on a $3 \times 4$ grid of initial positions that span a range of 12cm $\times$ 85.6cm, for a total of 12 trials per tool. We plot the successes and failures for each tool according to geometric position in Figure 10. We can see that the control policy is able to use each tool to solve the task for several initializations, but each tool is specialized for particular regions.

| Hyperparameter | Value |
| --- | --- |
| Tool Position Init. | (20, 10) |
| Control Steps Per Action | 1 |
| Max Episode Steps | 150 |
| Slack Reward | -0.001 |
| Tool Length Ratio | (-0.5, 0.5) |
| Tool Length Init. | (2.0, 2.0, 2.0) |
| Tool Angle Init. | (0.0, 0.0, 0.0) |
| Tool Angle Ratio | (-1.0, 1.0) |
| Tool Angle Scale | 90.0 |
| Control GNN | (64, 64, 64) |
| Control Index MLP | (128, 128) |
| Design GNN | (64, 64, 64) |
| Design Index MLP | (128, 128) |
| Control Log Std. | -1.0 |
| Design Log Std. | -2.3 |
| Fix Design & Control Std. | True |
| Policy Learning Rate | 2e-5 |
| Entropy $\beta$ | 0.01 |
| Value Learning Rate | 1e-4 |
| KL Divergence Threshold | 0.005 |
| Batch Size | 50000 |
| Minibatch Size | 2000 |
| PPO Steps Per Batch | 10 |

Table 5: Hyperparameters used for our framework on the push task.

| Hyperparameter | Value |
| --- | --- |
| Tool Position Init. | (20, 10) |
| Control Steps Per Action | 1 |
| Max Episode Steps | 150 |
| Slack Reward | -0.001 |
| Tool Length Ratio | (-0.5, 2.0) |
| Tool Length Init. | (2.0, 1.0, 1.0) |
| Tool Angle Init. | (0.0, 0.0, 0.0) |
| Tool Angle Ratio | (-1.0, 1.0) |
| Tool Angle Scale | 60.0 |
| Control GNN | (64, 64, 64) |
| Control Index MLP | (128, 128) |
| Design GNN | (64, 64, 64) |
| Design Index MLP | (128, 128) |
| Control Log Std. | 0.0 |
| Design Log Std. | 0.0 |
| Fix Design & Control Std. | True |
| Policy Learning Rate | 2e-5 |
| Entropy $\beta$ | 0.01 |
| Value Learning Rate | 1e-4 |
| KL Divergence Threshold | 0.002 |
| Batch Size | 50000 |
| Minibatch Size | 2000 |
| PPO Steps Per Batch | 10 |

Table 6: Hyperparameters used for our framework on the catch balls task.

| Hyperparameter | Value |
| --- | --- |
| Tool Position Init. | (15, 10) |
| Control Steps Per Action | 5 |
| Max Episode Steps | 30 |
| Slack Reward | -0.001 |
| Tool Length Ratio | (-0.7, 0.2) |
| Tool Length Init. | (6.0, 3.0, 3.0) |
| Tool Angle Init. | (0.0, 0.0, 0.0) |
| Tool Angle Ratio | (-0.1, 0.7) |
| Tool Angle Scale | 90.0 |
| Control GNN | (64, 64, 64) |
| Control Index MLP | (128, 128) |
| Design GNN | (64, 64, 64) |
| Design Index MLP | (128, 128) |
| Control Log Std. | 0.0 |
| Design Log Std. | 0.0 |
| Fix Design & Control Std. | True |
| Policy Learning Rate | 2e-5 |
| Entropy $\beta$ | 0.01 |
| Value Learning Rate | 3e-4 |
| KL Divergence Threshold | 0.1 |
| Batch Size | 50000 |
| Minibatch Size | 2000 |
| PPO Steps Per Batch | 10 |

Table 7: Hyperparameters used for our framework on the `scoop` task.

| Hyperparameter | Value |
| --- | --- |
| Tool Position Init. | (0.0, 0.5, 0.02) |
| Control Steps Per Action | 10 |
| Max Episode Steps | 100 |
| Slack Reward | -0.001 |
| Success Reward | 10.0 |
| Box Dimensions Min | (0.005, 0.05, 0.005) |
| Box Dimensions Max | (0.015, 0.1, 0.02) |
| Tool Angle Min | (-90.0, -90.0, -90.0) |
| Tool Angle Max | (90.0, 90.0, 90.0) |
| Control Action Min | (-1.0, -1.0, -1.0, -0.2, -0.2, -0.2) |
| Control Action Max | (1.0, 1.0, 1.0, 0.2, 0.2, 0.2) |
| Control Action Scale | 0.1 |
| Control GNN | (128, 128, 128) |
| Control Index MLP | (128, 128) |
| Design GNN | (128, 128, 128) |
| Design Index MLP | (128, 128) |
| Control Log Std. | 0.0 |
| Design Log Std. | 0.0 |
| Fix Design & Control Std. | False |
| Policy Learning Rate | 1e-4 |
| Entropy $\beta$ | 0.0 |
| Value Learning Rate | 3e-4 |
| KL Divergence Threshold | 0.5 |
| Batch Size | 50000 |
| Minibatch Size | 2000 |
| PPO Steps Per Batch | 10 |

Table 8: Hyperparameters used for our framework on the `fetch cube` task.

| Hyperparameter | Value |
|---|---|
| Tool Position Init. | (0.0, 1.2, 0.05) |
| Control Steps Per Action | 10 |
| Max Episode Steps | 150 |
| Slack Reward | -0.001 |
| Success Reward | 10.0 |
| Box Dimensions Min | (0.005, 0.02, 0.01) |
| Box Dimensions Max | (0.01, 0.1, 0.03) |
| Tool Angle Min | (-30.0, -30.0, -30.0) |
| Tool Angle Max | (30.0, 30.0, 30.0) |
| Control Action Min | (-1.0, -1.0, -1.0, -1.57, -1.57, -1.57) |
| Control Action Max | (1.0, 1.0, 1.0, 1.57, 1.57, 1.57) |
| Control Actioin Scale | 0.1 |
| Control GNN | (128, 128, 128) |
| Control Index MLP | (128, 128) |
| Design GNN | (128, 128, 128) |
| Design Index MLP | (128, 128) |
| Control Log Std. | 0.0 |
| Design Log Std. | -1.0 |
| Fix Design & Control Std. | True |
| Policy Learning Rate | 2e-5 |
| Entropy $\beta$ | 0.01 |
| Value Learning Rate | 3e-4 |
| KL Divergence Threshold | 0.5 |
| Batch Size | 50000 |
| Minibatch Size | 2000 |
| PPO Steps Per Batch | 5 |

Table 9: Hyperparameters used for our framework on the `lift cup` task.

| Hyperparameter | Value |
|---|---|
| Tool Position Init. | (0.0, 0.05, 0.1) |
| Control Steps Per Action | 10 |
| Max Episode Steps | 100 |
| Slack Reward | -0.001 |
| Success Reward | 10.0 |
| Box Dimensions Min | (0.04, 0.005, 0.02) |
| Box Dimensions Max | (0.08, 0.005, 0.05) |
| Tool Angle Min | (-15.0, -15.0, -15.0) |
| Tool Angle Max | (15.0, 15.0, 15.0) |
| Control Action Min | (-1.0, -1.0, -1.0, -1.57, -1.57, -1.57) |
| Control Action Max | (1.0, 1.0, 1.0, 1.57, 1.57, 1.57) |
| Control Action Scale | 0.05 |
| Control GNN | (128, 128, 128) |
| Control Index MLP | (128, 128) |
| Design GNN | (128, 128, 128) |
| Design Index MLP | (128, 128) |
| Control Log Std. | 0.0 |
| Design Log Std. | 0.0 |
| Fix Design & Control Std. | False |
| Policy Learning Rate | 1e-4 |
| Entropy $\beta$ | 0.01 |
| Value Learning Rate | 3e-4 |
| KL Divergence Threshold | 0.5 |
| Batch Size | 20000 |
| Minibatch Size | 2000 |
| PPO Steps Per Batch | 5 |

Table 10: Hyperparameters used for our framework on the 3D `scoop` task.

