# OpenReview forum: "Learning to Design and Use Tools for Robotic Manipulation"
_robot-learning.org/CoRL/2023/Conference — CoRL 2023 Poster_

### Official Review · Reviewer_1xsZ · 2023-07-13

**Confidence:** 5
**Originality:** Very Good
**Technical Quality:** Very Good
**Clarity Of Presentation:** Excellent
**Impact:** 4

**Recommendation:**

Strong Accept: I recommend accepting the paper and will argue for my recommendation even if other reviewers hold a different opinion.

**Review:**

This work provides a novel task, which is a possible way of overcoming the morphology limitations of robot. It brings significant insight of how a embodied agent can learn to expand its physical ability to manipulate the world. The method and experiments are clearly described but some parts of the experiment result are missing.
1. Strength:
- A novel task that demonstrates how the embodied agent can learn to build tools to expand its morphology ability
- The generalization is well demonstrated in the experiments.
- The proposed method is straightforward and intuitive.
2. Weakness:
- Different parameterization configurations limit the ability of how the algorithm can generalize to new types of tasks.


**Quality Of The Limitations Section:**

Limitations are addressed clearly

**Questions For Rebuttal:**

1. If the degree of freedom provided to the neural network differs from task to task, this seems like an over-simplification of the proposed task. Or is it too difficult currently to use a more general parameterization such as generating a 2D mask to represent the shape of a 2D tool?
2. How is the performance of Scoop (3D)? I cannot find the experiment results in terms of Scoop (3D) task.
3. How to decide the position to pick the tool after they are fabricated in the real world?
4. The author mentions using a two-stage strategy to avoid collision with the overhang. Does this change the output of the control policy?


**Robotics Focus:**

Sufficient demonstration on hardware

**Summary Of Paper:**

This work presents a novel task that unlocks the ability of a embodied agent to design and use tools. The authors propose a two-stage training pipeline consisting of a designer network and a controller network. They train two networks jointly using PPO and implement the network using graph neural network. In the experiment session, the authors test 5 tasks in both simulation and real-world Franka Panda arm setup. To further study the performance of the proposed framework, the authors investigate the sample efficiency and the ability to generalize to unseen tasks by zero-shot generalization and finetuning.

**Summary Of Recommendation:**

This work proposes a novel task that allows the embodied agent to expand its ability by designing task-dependent tool. The idea and methods are described clearly. The limitations are well addressed. Thus, I would recommend a strong accept for this work.

---

> ### Author Response · Authors · 2023-08-10
> **Response to 1xsZ**
>
> Thank you for your thoughtful review and positive feedback. We answer your questions below:
>
> > Degrees of freedom of the tool parameterization?
>
> This is a great question – we have in early stages of the project explored precisely the strategy of generating a 2D mask to represent the shape of a 2D tool. Our framework can also learn to design tools of this parameterization, but we found to our surprise that the simple chain link parameterization is able to solve a wide range of tasks, and also result in improved sample efficiency compared to more complex tool parameterizations.
>
> > Performance of Scoop (3D)?
>
> The training curves for the Scoop (3D) task can be found in Figure 4 (bottom right). We did not evaluate the real-world performance of Scoop (3D) due to perception challenges such as significant occlusion of the objects by the robot arm and by each other.
>
> > How to decide the position to pick the tool after they are fabricated in the real world?
>
> In each task, the starting position of the robot’s end effector (and thus the tool) is defined as part of the environment. For the real world experiments, we mirror this scenario as closely as possible by having the robot grasp the tool and move to the defined starting position.
>
> > Does [the two-stage strategy of the robot performing the fetch cube task] change the output of the control policy?
>
> We do not explicitly assign or enforce a two-stage manipulation strategy for any task; rather, our observation of the two-stage strategy is a qualitative description of some intriguing behavior exhibited by the learned control policy.
>
> We hope these responses help and are happy to continue to discuss any further questions!

---

> > ### Comment · Reviewer_1xsZ · 2023-08-11
> > **Discussion and additional questions**
> >
> > Thank for the answers. There is one further question related to the tool parameterization.
> >
> > > *Degrees of freedom of the tool parameterization?*
> >
> > Sample efficiency is a good point. It still confuses me that the lifting cup task does not use a tool which can not be parameterized by a chain link. (also in scoop 3D task) It is a tree structure. Does this means you need to design the parameterization (e.g. number of links, the topological structure) for each task?

---

> > > ### Author Response · Authors · 2023-08-14
> > > **Response to additional questions**
> > >
> > > > Sample efficiency is a good point. It still confuses me that the lifting cup task does not use a tool which can not be parameterized by a chain link. (also in scoop 3D task) It is a tree structure. Does this means you need to design the parameterization (e.g. number of links, the topological structure) for each task?
> > >
> > > Thank you for the follow up question. Our approach is agnostic to different tool design parameterizations, and could allow the policy to freely choose the number of links and the topological structure to attach those links. However, as we noted in Section 4 of our paper, when the design space is large and contains many designs that are unhelpful for solving any task, the reward signal becomes very sparse, as the joint design and control learning problem has an extremely large search space. Therefore, as a first step towards tackling this problem, we provide the agent with a minimal human-designed tool parameterization to start with for each task, and give the agent the complete freedom to design the best tool within that parameterization that solves the task at hand. As indicated by the performance of baseline methods, these parameterizations alone are not sufficient for task success. Our method's improved learning of optimal design parameters is critical.

---

### Official Review · Reviewer_2UAy · 2023-07-14

**Confidence:** 5
**Originality:** Good
**Technical Quality:** Poor
**Clarity Of Presentation:** Very Good
**Impact:** 3

**Recommendation:**

Weak Accept: I recommend accepting the paper, but will not argue for my recommendation if the majority of other reviewers have a different opinion.

**Review:**

The paper presents a framework for learning tool design and use in robotics, specifically for manipulation tasks. The framework consists of a two-phase Markov Decision Process (MDP) representing the design and control phases of the agent's environment. A designer and controller policies are jointly trained using interactions with the environment in both phases.

In the design phase, the designer policy selects the parameters for the tool that will be used throughout the episode. The environment state consists of task observations, such as object positions and velocities, as well as the robot's end-effector position, if applicable. In the control phase, the controller policy generates actions that actuate the previously designed tool. The control state includes the task observations and the design action.

This is a very interesting area, and the setting is also intriguing, where it asks the question of the best tool design for a given task.
However, the whole setup feels like it is more targeting designing experiments (for "inventing" the best tools). This is an area that is being tackled in drug development [1] and as a sequential optimization problem [2]. It is an interesting problem. However, it can be an entirely decoupled problem with nothing to do with the robot arm. The paper uses two different policies \pi_D and \pi_C to model the "designer" and "controller" separately, and it was also found to be the better architecture than the proposed model with shared-arch (i.e., a unified one) (figure 4). The authors argue that:

> The ablation shows the importance of learning separate designer and controller policies. (lines 225-226)

However, what is the size of the network parameters? Does the "shared arch" essentially consists of half the trainable parameters count than the "designer"-"controller" counterpart? Is the "worse performance" contributed by having less trainable parameters? It would also be informative to have the corresponding training/inference time for each comparing model to assess the proposed model's actual performance correctly.

The fact that the solving of the whole robot control problem is decoupled into "designer" and "controller" models suggested that the problem is just being solved separately (and just so happened to be set to optimize in the same iteration step). A possible way for the author to argue their approach is to use some of the more advanced RL algorithms, like TD3, SAC, PPO, Rainbow DQN, etc., as a comparison as opposed to other ad-hoc models. Instead of using a two-stage "designer" / "controller" optimization step, other models can simply be optimizing all parameters jointly (both to control the arm and the design parameters of the tool).


The paper selects a design space for tools composed of rigid links, sufficient for manipulation tasks, and easily deployed in the real world. The policies are trained using proximal policy optimization (PPO), and a graph neural network (GNN) policy and value function architecture is adopted. Randomized goals are sampled from the environment during training in goal-conditioned environments. However, it is very hard to understand the result of the product from the "designer." A better way to illustrate the designer's strength is to directly compare the final tools that each model created for some given tasks (to see why certain tool designs are superior to others). Concerning the videos, perhaps a video to see the evolution of the designed tool over the optimization epoch would also be highly useful to see how the tool evolves over the training horizon.


I also have high doubts about the method used to evaluate the control of the robot. In lines 177-178, it was mentioned that 3D end-effector positions were used to control the arm. Based on the methodology in the paper and the video shown in the supplementary, the pose of the arm never changes. Despite the one short-coming of controlling the robot in world-space instead of configuration space (which is already limiting the expressiveness of controlling the robot), shouldn't the robot at least be controlled with the full SE(3) of the end-effector?

In addition, I need clarification on why the 2D scenario uses velocity control while the 3D scenario only uses position control. (Both 2D & 3D scenarios have "Scoop" action, which seems like they should be using a similary method to allow for a direct comparison?)

Resource constraints are also considered, and a trade-off between design material cost and control energy consumption is introduced through an auxiliary reward. This reward adjusts the task reward at each environment step based on the utilization of tool components and control velocity.

Lastly, based on the tool design figure given in Figure 2 (which consists of [ l_{1,...,3}, \theta_{1, 2} ] ), I cannot see how the "designer" can possibly come up with the design shown in Figure 9 (bottom row). None of the parameters control the thickness of the "stick", no? Unless each task has its own specific default thickness across the "sticks"? However, if that is the case, that would imply that the author injects bias towards each task (instead of being a unified auto-tool-design framework across tasks).


[1] Réda, C., Kaufmann, E., & Delahaye-Duriez, A. (2020). Machine learning applications in drug development. Computational and structural biotechnology journal, 18, 241-252.
[2] Blau, T., Bonilla, E. V., Chades, I., & Dezfouli, A. (2022, June). Optimizing sequential experimental design with deep reinforcement learning. In International Conference on Machine Learning (pp. 2107-2128). PMLR.

**Quality Of The Limitations Section:**

Limitations are not well addressed

**Questions For Rebuttal:**

Please address questions raised as a result of the review given.

Namely:
- Experimental details on model size, architecture, and comparison to SOTA RL models with one unified model.
- Visualisation of the evolution and questions on the tool design
- Questions on the controlling aspect of the robot

**Robotics Focus:**

Irrelevant to robotics

**Summary Of Paper:**

The article discusses the potential of using objects from the environment as tools to accomplish tasks beyond the capabilities of humans and animals. It suggests that robots can also benefit from tool use by combining deep learning techniques to optimize both their physical design and control.

This paper proposes a "designer policy" approach, where a policy conditioned on task information generates tool designs to solve the task, while a separate controller policy performs manipulation using these tools.

The authors present a reinforcement learning framework that demonstrates improved sample efficiency compared to previous methods in scenarios with multiple goals or variants. It also showcases the ability to handle new tasks through zero-shot interpolation or fine-tuning, enabling trade-offs between design and control complexity.

**Summary Of Recommendation:**

I think the idea is interesting, but the approach used to tackle the problem seems to be standalone (does not seem to be tackling with robotics in mind). There are also questions on the approach used to evaluate the proposed method (needs more experiments and details)

-----------------------------

**Post rebuttal**

The author had addressed some of my concerns on the contribution and the lack of experimental details / highlights of this work's contribution. I will raise my rating.

---

> ### Author Response · Authors · 2023-08-10
> **Response to 2UAy (1/2)**
>
> Thank you for your thoughtful feedback that has helped us improve our work. We amended the paper with the requested experimental details and have added additional visualizations on the following website: https://robotic-tool-design.github.io/rebuttal.html. We answer your questions below:
>
> > The setup feels like it is targeting designing experiments for “inventing” the best tools, which is “an area that is being tackled in drug development [1] and as a sequential optimization problem [2]”.
>
> We would like to clarify that our problem setting is fundamentally different from experimental design. To our knowledge, in the work of Blau et al [2], the aim is to learn a policy that, at deployment time, selects which experiment to run next based on the history of experiments in order to optimize the information gain from the experiment. In contrast, our aim is to learn a policy that, at deployment time, directly outputs the best tool design for the given scenario, as well as a control policy that allows a robot to manipulate it. Therefore two major distinctions are that: 1) our method experiments with “inventing” the best tools during training time, rather than during deployment, and 2) the reward signal for the success of designs comes indirectly from the success of a jointly learned manipulation policy at solving the task, rather than based on information gain.
>
> > “[The problem] can be an entirely decoupled problem with nothing to do with the robot arm”
>
> While the concept of jointly learning design and control policies via reinforcement learning can indeed be applied to settings outside of robotics, our work focuses on developing a specific framework for robotic tool use. The robot arm is integral to our problem setting, as the learned control policies directly output the motor commands of a robot arm. Our robotics-specific design decisions and contributions include:
>
> Developing a rigid tool parameterization for robotic manipulation tasks
> New simulated robotic manipulation environments for this problem setting
> An adjustable power/material tradeoff parameter
> Real-world hardware evaluations for physically fabricated tools.
>
> > Number of network parameters for the “shared arch” ablation:
>
> Thank you for raising this important point. We control for the number of network parameters in the “shared arch” ablation – notably, we used MLP policies implemented in Stable Baselines 3 (Raffin et al.) and ensure that the number of trainable network parameters is either the same or strictly larger than in our method across all tasks. We have added a note of this to appendix B.2.
>
>  > Training/inference time for each model:
>
> The training/inference time for each model on the Fetch Cube task is as follows:
>
> |                | Ours         | Ours (shared arch) | CMA-RL       | HWasP-minimal/Ha 2018 | CMA-ES                              |
> |----------------|--------------|--------------------|--------------|-----------------------|-------------------------------------|
> | Training time  | 2.121 x 10^2 | 7.287 x 10^2       | 1.180 x 10^3 | 2.982 x 10^2          | 3.456 x 10^2                        |
> | Inference time | 1.68 x 10^-3 | 5.24 x 10^-4       | 5.32 x 10^-4 | 1.75 x 10^-3          | N/A (trajectory directly optimized) |
>
> The training time represents the wall clock time in minutes needed to train the model for 1 x 10^7 environment steps. The inference time denotes the wall clock time in seconds needed to perform one forward pass through the model. These results are generated using a NVIDIA Titan RTX GPU and an Intel Xeon Gold 5220 CPU.
>
> All methods, including our own, are sufficiently performant to enable real-time control on a real robot.
> We have added this information to Appendix B.3.
>
> > “[That] the whole robot control problem is decoupled into "designer" and "controller" models suggests that the problem is just being solved separately.”
>
> To clarify, while we decompose the policies into “designer” and “controller” models, the design and control problems cannot be solved separately as they are highly interdependent. This is because the designer policy only receives rewards when the controller policy is able to successfully use the tool to solve the task. Conversely, if the design policy is performing poorly, the control policy will be unable to learn as the task may be unsolvable without a suitably designed tool, removing any optimization signal for the design policy.

---

> ### Author Response · Authors · 2023-08-10
> **Response to 2UAy (2/2)**
>
>
> >  “A possible way for the author to argue their approach is to use some of the more advanced RL algorithms …  as a comparison as opposed to other ad-hoc models… Other models can simply be optimizing all parameters jointly (both to control the arm and the design parameters of the tool)”
>
> We believe that our “shared arch” experiment is representative of the suggested comparison, as it uses a single policy to output both designs and control actions, and also optimizes all parameters jointly with PPO. (Our method also uses PPO for the policy learning step.) Please let us know if this is the comparison you have in mind – if not, we would be happy to look into the suggested baseline.
>
> > Visualizing the evolution of the designed tools over the epochs
>
> This is an excellent suggestion. We added visualizations of designed tools over the course of training for several tasks to our webpage https://robotic-tool-design.github.io/rebuttal.html. We will include these types of visualizations in future project website versions and the project video.
>
> > The method used to control the robot – shouldn’t the robot be at least controlled with the full SE(3) of the end-effector?
>
> We limit the action space of the robot for some of the 3D environments to reduce exploration challenges, which are not the main focus of our study. Note that the Scoop3D environment performs full SE(3) control, and the other environments can be readily adapted to do so as well, at the cost of increased environment interactions during training as the action space becomes higher-dimensional.
>
> > The 2D scenario uses velocity control while the 3D scenario uses position control
>
> The choice to use velocity control for the 2D scenario and position control in the 3D scenario was a design choice based on the availability of simulation functions available in the respective simulators (Box2D and Pybullet). Our method is agnostic to the choice of action representation as a benefit of learning end-to-end via reinforcement learning.
>
> > Tool design in Figure 2 vs Figure 9
>
> We apologize for the confusion – the design shown in Figure 2 illustrates the parameters for the tools in the 2D environments, which does not include the tool link thickness. In the 3D environments, the tool parameterizations do include dimensions of the tool chain links. These dimensions are automatically learned.
>
> > Visualizing final designed tools
>
> This is also a great suggestion – we are in the process of producing these visualizations and wanted to begin the discussion as early as possible, but we will add these during the rebuttal period.
>
> Thank you again for your review and helpful feedback! We look forward to continuing the discussion.

---

> > ### Comment · Reviewer_2UAy · 2023-08-13
> > **response to author**
> >
> > Thank you for the in-depth and detailed responses.
> >
> > The additional technical details and clarifications are useful and are much appreciated. The following are some responses to the helpful comments provided by the authors
> >
> > > [...] the work of Blau et al [2], the aim is to learn a policy that, at deployment time, selects which experiment to run next based on the history of experiments [...]. In contrast, our aim is to learn a policy that, at deployment time, directly outputs the best tool design for the given scenario, as well as a control policy that allows a robot to manipulate it [...]
> >
> > Thank you for pointing out the original intention of this work, as that was not clear that this work aims to directly output the best tool for the job at deployment time without retraining.
> >
> > However, I think based on the evaluation approach in the paper, the end-results are still the same as the former case (of experiment design), no? As in, it seems like a separate policy is still needed to train for all separate scenarios (i.e., the **Push**, **Catch balls**, **Scoop**, etc. all require a separate training loop for getting a new policy). It's not like given a novel environment, the proposing method can directly output a "best tool" without yet another bunch of training episode. Or am I misunderstanding something from your argument?
> >
> > *Or, say, in the weaker sense of directly producing the best tool:* under the same environment (e.g. **Fetch cube** or the new **Open drawer** scenario), if we randomise the location of the object (e.g. drawer) while retaining the nature of the task; does the tool that it produces differ based on the random configuration? Or does it always try to use more-or-less the same tool configuration to solve for the given task? I don't think I've seen any discussion in this regard in the text. However, if it is the latter case, then it would mean that the learned policy is not **adaptive** in the sense that it can "just" produce the best tool at deployment time.
> >
> > > While the concept of jointly learning design and control policies via reinforcement learning can indeed be applied to settings outside of robotics, our work focuses on developing a specific framework for robotic tool use. The robot arm is integral to our problem setting, as the learned control policies directly output the motor commands of a robot arm. [...]
> >
> > My main concern is that, this problem formulation feels very much like under the realm of Bayesian Optimisation.  In the manuscript, the term BO was mentioned once but without any elaboration. Perhaps the author can argue on this point more to highlight the differences between the two formulation.
> >
> > Tool design is not as much of a "control problem" (which RL for sure is good for), as it's more just like parameter optimisation. I'm not sure how the underlying MDP formulation in RL theory would plays a role here in tool design.
> >
> > > Tool design in Figure 2 vs Figure 9 [...] We apologize for the confusion [...] the tool parameterizations do include dimensions of the tool chain links. These dimensions are automatically learned
> >
> > Thank you for the clarification on the parameterizations. However, please response to my previous concern on the approach of setting the parameterisation process. Specifically, my original comment:
> >
> > > Lastly, based on the tool design [...] how the "designer" can possibly come up with the design [...] However, if [each task has its own parameters], that would imply that the author injects bias towards each task (instead of being a unified auto-tool-design framework across tasks).
> >
> > Upon further inspection of the supplementary material, I can see that there are different parameterisations in different tasks. Despite the 2D/3D differences, there are still different parameterisation within the 3D tasks. (Fetch cub has n=11, Lift cup has n=5).
> >
> > Firstly, please provide details on what those parameters mean (e.g., similar to how you specify the vector of theta/length in your original manuscript).
> >
> > Secondly, relating to my previous concern about **injecting author bias**, why do different tasks require different parameterisation if they use the same basic tool template to attempt the given task? And how would that relate to your previous response to the claim of: **"at deployment time, directly outputs the best tool design for the given scenario"**?
> >
> > > We added visualizations of designed tools over the course of training for several tasks
> >
> > Thanks for the additional visualisation. However, I was thinking more along the line of only focusing on the tool with a fixed frame (something like this topology morphing: https://upload.wikimedia.org/wikipedia/commons/2/26/Mug_and_Torus_morph.gif). I think that would be a great way to visualise the tool deforming across episodes.
> >
> > ----------------------
> >
> > Every other response that is not directly addressed above (e.g. the added elaboration, clarifications, information on training/inference time, etc.) are noted and very much appreciated.

---

> > > ### Author Response · Authors · 2023-08-15
> > > **Follow-up to 2UAy (1/2)**
> > >
> > > Thank you for your follow-up questions. We very much appreciate the continued discussion, and hope to clarify and answer your questions as follows:
> > >
> > > > It seems like a separate policy is still needed to train for all separate scenarios... In the weaker sense of directly producing the best tool: under the same environment, if we randomise the location of the object... does the tool that it produces differ based on the random configuration?
> > >
> > > This is an excellent question. It is correct that we train a separate policy for each scenario (for example, **push**, **catch balls**, etc.) However, this policy **does** produce different tools based on the random configuration, for example, the position of the cube in **fetch cube**, or the number of desired scooped objects in **scoop**. This is critical because different tool designs can be required for different scenarios. The visualizations at the bottom of the project webpage (https://robotic-tool-design.github.io/) illustrate a few examples:
> > >
> > > - For **fetch cube**, in the leftmost example, the policy designs a longer tool that allows the robot to reach a cube that is further below the obstacle. In the two examples on the right, the produced tool is shorter and more hook-like, allowing the cube to be retrieved more quickly.
> > > - For **lift cup**, the policy learns to deesigns tools for different cup sizes such that they can be wide enough to fit the base of the cup but ensure stable contact.
> > > - For **scoop (3D)**, the policy is instructed to design tools for scooping up exactly one, four, and seven cubes (from left to right). The policy learns to design the tool to fit the desired number of cubes but not larger, which would increase the likelihood of scooping extra cubes.
> > >
> > > Figure 5 in the text also shows examples of these different tools for the **push (2D)** task as the goal configuration varies – in that task, the policy reasonably learns to include a face of the tool roughly perpendicular to the pushing direction.
> > >
> > > One critical advantage of our method, as we discuss in Section 5.2 of the manuscript, is how our learned policies can generalize to task configurations unseen during training. In particular we show how we can apply the design and control policies learned in the **fetch cube** task to random configurations **unseen during training**.
> > >
> > > Thus, our learned policies are indeed **adaptive**, and this is the main motivation for our work – to our knowledge, our work is the first to enable **configuration-specific tools to be directly designed based on the configuration at hand**. We will revise the manuscript to make it more clear that the tools used to solve different task configurations are different.
> > >
> > > > Relation to Bayesian Optimisation.
> > >
> > > Thank you for raising this point. While BO can be applied to this problem, it is not obvious how it can be used to **adaptively** produce designs at deployment time.
> > >
> > > One prior Bayesian optimization approach to a joint design and control problem by Liao et al. [1] iterates between performing an outer-loop Bayesian optimization on the design parameters and inner-loop Bayesian optimization on control policy parameters. While this has shown the ability to perform co-design of design and control for a **single goal** (forward distance traveled), the entire optimization process must be performed again for every desired configuration (e.g. moving backwards). Furthermore, Liao et al. use an open loop control parameterization to minimize the number of control parameters. In contrast, as discussed above, our method learns closed-loop policies that can produce tool designs and control tools by directly conducting a forward pass.
> > >
> > > Another approach is to use BO for black-box outer-loop design optimization while learning control policies in the inner loop via RL. We presented a similar baseline in Figure 4 that uses another black-box optimizer, CMA-ES, as the outer-loop design optimization strategy instead of BO (“CMA-RL”). We found that this was unable to achieve competitive performance, and again, with this approach it is unclear how to design a tool for a novel configuration without re-running optimization.
> > >
> > > We will expand on the discussion of BO in the manuscript.
> > >
> > > > Tool design is not as much of a "control problem" (which RL for sure is good for), as it's more just like parameter optimisation. I'm not sure how the underlying MDP formulation in RL theory would play a role here in tool design.
> > >
> > > Formulating the problem as a combined design/control (“two-phase”) MDP allows us to naturally introduce the adaptive designer as a neural network policy that takes as input the environment configuration and outputs tool design parameters. This is in contrast to approaches, for example BO, that optimize a single design parameter vector. It also allows for control-phase experiences to be shared across designs, improving sample efficiency.
> > >
> > > ---
> > >
> > > [1]: “Data-efficient Learning of Morphology and Controller for a Microrobot”, Liao et al. 2019

---

> > > ### Author Response · Authors · 2023-08-15
> > > **Follow-up to 2UAy (2/2)**
> > >
> > > > The parameterization:
> > >
> > > > Firstly, details on what those parameters mean
> > >
> > > We specify the meanings of the parameters for each task below:
> > >
> > > - *push (2D), catch balls (2D), scoop (2D)*: the tool is a 3-link chain represented by a vector $[l_1, l_2, l_3, \theta_1, \theta_2]$ where $l$ represents each link length and $\theta$ represents the relative angle between the links, as shown in Figure 2.
> > > - *fetch cube (3D), open cabinet (3D), open drawer (3D)*: The tool is a three-link chain represented by a vector $[l_1, h_1, w_1, l_2, h_2, w_2, l_3, h_3, w_3, \theta_1, \theta_2]$ where $l_i, w_i, h_i$ represent the length, width, and height of each box. $\theta_1$ and $\theta_2$ represent the relative angles between the links along the x axis.
> > > - *lift cup (3D)*: The tool is a four-link fork with two prongs parameterized by a vector $[d, \theta, w, l, h]$, which represent the separation, tilt angle, width, length, and height of the prongs respectively. The same parameters are applied to both prongs to maintain symmetry.
> > > - *scoop (3D)*:  The tool is a six-link scoop composed of a rectangular bottom plate and four rectangular side plates attached to each side of the bottom plate. The parameterization is a vector $[w, l, h_1, \theta_1, h_2, \theta_2, h_3, \theta_3, h_3, \theta_4, h_4]$ representing the width and length of the bottom plate and height and relative angle of each side plate.
> > >
> > > As you mentioned, these definitions are currently in Appendix A. We have expanded their descriptions to include the above.
> > >
> > > >  Why do different tasks require different parameterisation if they use the same basic tool template to attempt the given task?
> > >
> > > As described above, while some of our tasks share tool templates (for example all of the 2D tasks), in general our tasks use different tool templates. We apologize for the confusion.
> > >
> > > The reason is that as noted in Section 4 of the paper, when the design space is large and contains many designs that are unhelpful for solving any task, the reward signal becomes very sparse, as the joint design and control learning problem has an extremely large search space. Therefore, as a first step towards tackling this problem, we provide the agent with a minimal human-designed tool parameterization to start with for each task, and give the agent the complete freedom to design the best tool within that parameterization that solves the task at hand. As indicated by the performance of baseline methods in Figure 4, these parameterizations alone are not sufficient for task success. That our method learns to select the appropriate design parameters is critical.
> > >
> > > > And how would that relate to your previous response to the claim of: "at deployment time, directly outputs the best tool design for the given scenario"?
> > >
> > > To relate it to our previous claim, in this work our method enables the agent to output the best tool design for a given **task configuration**, for example the position of the cube or number of desired objects to scoop, but we do not yet demonstrate its ability to generalize across scenarios (e.g. from fetch cube to open drawer), which is an exciting direction for future work. We will make this more clear in the paper text.
> > >
> > > > Visualization
> > >
> > > Thank you for the suggestion – we agree that this is a great way to visualize the tools’ evolution across episodes, and have added it for several of the tasks in the new Figure R1 of our webpage (https://robotic-tool-design.github.io/rebuttal). We will update our project page to use this visualization style for all tasks.
> > >
> > > > Visualizing final designed tools
> > >
> > > As previously suggested, we have added visualizations of final designed tools for the fetch cube task in Figure R2 of our webpage (https://robotic-tool-design.github.io/rebuttal).. We will continue to add them for additional tasks for the remainder of the rebuttal stage.
> > >
> > > We hope that these responses can help clarify your questions and look forward to continued discussion. We sincerely appreciate your engagement; your feedback has been very helpful for us in improving the paper! Please let us know if you have any additional questions or concerns.

---

### Official Review · Reviewer_RS5q · 2023-07-19

**Confidence:** 4
**Originality:** Fair
**Technical Quality:** Good
**Clarity Of Presentation:** Good
**Impact:** 2

**Recommendation:**

Weak Accept: I recommend accepting the paper, but will not argue for my recommendation if the majority of other reviewers have a different opinion.

**Review:**

Strengths:

The paper addresses an interesting problem -- robotic tool using and design. And the paper tries to solve it by jointly optimizing the design policy and control policy. The presentation of the method is clear and straightforward. Figures clearly express the algorithm step by step. There are sufficient quantitative results, baseline comparison, and real-world experiments.

Weaknesses and suggestions:

1) The parametrized design morphology for tools is relatively simple and it simplifies the problems as well. Is there a more general way to describe the tool's morphology during design? (like using some 3D mesh)
2) In line 222, "It does so by producing specialized tools to solve each task instance, while the other methods optimize for a single design across all task variations. " The 'single design across all task variations' is confusing, why do baselines only optimize single design across all?
3) In figure5, though the goals changed, I do not understand why the final tool design pattern looks like this. If those goals occur uniformly during training, the output tool pattern should be similar to a parallel gripper, please provide more insights here.
4) Since your tool design is mainly based on geometry and morphology, the lack of considering the tool's stability/ applied force should also be mentioned in the limitation part.

**Quality Of The Limitations Section:**

Limitations are addressed clearly

**Questions For Rebuttal:**

As above

**Robotics Focus:**

Sufficient demonstration on hardware

**Summary Of Paper:**

The paper describes a learning method for designing and using tools for robot manipulation. The proposed method could jointly optimize tool design policy and tool use policy for solving tasks that are constrained by the robot's own morphologies. They also show their framework is sample efficient and could be transferred in multi-goal settings in the real world as well.



**Summary Of Recommendation:**

In summary, this paper proposes a framework for --tool-designing and using with RL methods. Though more details and explanations are needed to be added to show the validity of the method, I am leaning toward accepting this work.

---

> ### Author Response · Authors · 2023-08-10
> **Response to RS5q**
>
> Thank you for your thorough review and suggestions that have helped us to improve the paper. We respond to each of the points below:
>
> > More general way to describe the tool morphology:
>
> This is a very good question – in our initial experiments we have tried more general tool morphology parameterizations, for example, in the 2D case, image or polygon-based representations. We found that these can also work and may be helpful for particular tasks. However, one lesson we learned through this work is that in a surprising number of cases, a chain-link representation as we use in the paper is sufficient to solve the task. For examples, see our newly added cabinet and drawer opening tasks at https://robotic-tool-design.github.io/rebuttal.html. It also results in a smaller, easier-to-optimize design space, decreasing the number of environment interactions required to solve the task.
>
> > Baselines optimizing single designs:
>
> To clarify, the baselines from prior work or existing joint optimization paradigms that we compare to were originally formulated to optimize a single set of design parameters along with the control policy. Thus we note this as one explanation for the improved performance of our method. We did not modify that aspect of the baselines.
>
> > Tool patterns in Figure 5:
>
> The tool shown in each box of Figure 5 is the tool outputted by the designer policy when presented with the goal shown in that box. The designer policy can design different tools depending on the goal. The tools appear hook-like with faces roughly parallel to the direction of the ball, which allows the agent to directly push the ball to stop at the goal in the smallest number of timesteps. There may be other feasible geometries besides those output by our method.
>
> > Stability and applied force:
>
> Thank you for raising this important point – our work indeed does not yet consider tool stability/applied force and we have expanded our limitations section in the manuscript to include this limitation.
>
> We hope that these responses have helped to answer your questions. Do you have any additional questions or concerns?

---

### Official Review · Reviewer_D5az · 2023-07-19

**Confidence:** 5
**Originality:** Good
**Technical Quality:** Good
**Clarity Of Presentation:** Good
**Impact:** 3

**Recommendation:**

Weak Reject: I recommend rejecting the paper, but will not argue for my recommendation if the majority of other reviewers have a different opinion.

**Review:**

Quality of the paper:
The paper is of interesting topic as we know that general manipulation (say in a single cell scenario or home robots) may require robots to be able to change their tools on the fly. With that respect I find the paper addresses the right problem which will be of interest in the years to come.Nevertheless, still the examples shown in this paper are of a toy-problem nature and I would expect (knowing that it is very challenging task to make it work) to see more realistic tasks. Scooping of reaching is TMHO quite a synthetic problems and they do not address the main problems of home robots or industrial robots.

Clarity:
The paper is written clearly.

Originality:
The algorithm to address the dual-nature tasks (design and control) if fairly straight forward. As the authors show in their experiments, they achieve SOTA performance. I believe that these results are authentic. I find the usage of GNN clever. Therefore, all in all this work has some original aspects.

Significance:
As mentioned in the quality section, I tend to asses that this work is significant, although in terms of realism we are not in realistic scenario yet.

Strengths:
The work is "closes the loop" from presenting the task, designing relevant solution, showing the validity in simulation, and implementing the solution on a real Franka Emika Panda robot. The design phase relied on RL is nice and valid approach TMHO.

Weaknesses:
The tasks are partially convincing and I find hybrid approaches of automation + AI more relevant. Also, I'm not sure that RL for control is the best way to handle the control phase. Grasping, reaching, with good perception may be achieved with low sample complexity where RL is not the right tool for control in grasping today (although there is abundant works in this direction by Sergey Levine and collaborators.).


**Quality Of The Limitations Section:**

Additional details required

**Questions For Rebuttal:**

1) Can you provide more tasks that can be solved with this approach? On real robot is the best, in simulation as well, or at least to mention more realistic scenarios where your approach may be relevant.

2) What are the resources that were used in order to train the system? Compute, time, etc.

3) What is the samples complexity on the robot to complete the tasks?

4) Was the transfer from simulation was in zero shot, one shot, few shot, finetunning?

5) Did the simulation was realistic, or generalization over the gap between simulation and real was possible?

**Robotics Focus:**

Sufficient demonstration on hardware

**Summary Of Paper:**

In this work, the authors present learning a robot that learns both to (1) design a tool and afterward (2) to use the tool in order to solve a manipulation task in hand. The manipulation tasks are pushing an object, catching a ball, scooping balls, fetching cubes, and lift cups. In training, they divide the training steps into two phases: "design" and "control". In the design phase the agent optimizes the tool at hand. In the control phase the agent learns how to use the tool. The agent maintain two separate policies for the the two phases. Some points were considered by the authors in order to enable efficient learning such as low dimensional design space but still enabling wide range of tasks, relevant algorithm (PPO) with relevant representation (GNN), and finally, good auxiliary reward. The algorithm was tested both in simulation and on real robot. Comparison to other tool design algorithms was done as well. The results showed that validity of the algorithm, even on unseen tasks.

**Summary Of Recommendation:**

I really thank the authors for the paper and appreciate their work. The decision is not trivial in this case. The work is fair and the problem is interesting. But I'm not sure the work addresses (with the current experiments, as detailed above) the bar of CoRL.

---

> ### Author Response · Authors · 2023-08-10
> **Response to D5az**
>
> Thank you for your thoughtful review and insightful comments. We have added additional experiments and answer your questions below:
>
> > Additional tasks and scenarios where this approach is relevant
>
> We have added additional tasks of learning design and control for cabinet and drawer opening at https://robotic-tool-design.github.io/rebuttal.html, and we also would like to clarify the real-world motivations for our existing tasks.
>
> We have added additional tasks, cabinet and drawer opening, to showcase how our method can be directly applicable to a home robotics setting. In these tasks the robot must open a cabinet above a kitchen counter, which prevents the robot from getting very close to it, or to open a low drawer, which is also out of reach. Similar tasks have been studied in numerous prior works [1, 2, 3] including previous works at CoRL, but care is usually taken to ensure that the objects are safely within reach of the robots. Our method successfully enables the robot to solve these tasks in more realistic scenarios, and results can be found on the linked website.
>
> Next, we provide examples of real-world applications that motivated the selection of our original tasks:
>
> - Fetching objects (fetch cube): retrieving fallen or misplaced objects e.g. underneath sofas, tables, or chairs in homes, or tight spaces like inside cars, enabling scene “resets” from states where objects are lost for continuous robotic learning settings.
> - Scooping (scoop 2D, 3D): manipulating granular materials such as rice, beans, cereals or measuring and transferring liquids for cooking.
> - Pushing (push): home robotics applications when robots are impeded by obstacles such as countertops, tables, beds; industrial applications to aligning and grouping objects for robotic packing.
> - Lifting objects (lift cup): Transporting objects that are too large for a parallel jaw gripper or for a suction gripper to stably grasp, for example, pots and pans, garbage cans, kitchen appliances; or risky to grasp directly, for example, in high temperature industrial welding and forging.
>
>
> We have updated Appendix A of the paper to expand the discussion of these applications.
>
> > Training resources?
>
> We train each of our policies using a single GPU (NVIDIA RTX 2080Ti or TITAN RTX) and 32 CPU cores. The total wall clock training time varies per environment from 2 hours for the Catch Balls environment to 24 hours for the Scoop(3D) environment. We have added these details to Appendix B.3. Thank you for raising this point!
>
> > Sample complexity to complete the tasks?
>
> The number of environment samples needed to achieve a reasonably good performance during training in simulation varies across different environments. Roughly:
> - Fetch cube: 2.0 x 10^7
> - Lift cup: 2.0 x 10^6
> - Scoop (3D): 1.0 x 10^7
> - Push: 7.5 x 10^6
> - Catch balls: 1.5 x 10^7
> - Scoop (2D): 1.0 x 10^7
>
>
> The policies learned in simulation were transferred zero-shot to the real world, so no additional samples are needed.
>
> > Transfer from simulation to real?
>
> The transfer from simulation to real was done in zero-shot. We use 3D-printing to convert the tools designed by the policy to physical objects, and then roll out the policy learned in simulation directly on the real robot, with no fine-tuning.
>
> > Realism of simulation & generalization gap:
>
> For the tasks that we considered in the paper, the simulation was sufficiently realistic for generalization from sim to real to be successful.
>
> > RL for the control phase:
>
> We agree that there are other approaches for grasping and reaching with improved sample efficiency, but we opt to use RL for the control phase in our framework because it makes our framework flexibly applicable to any task that can be specified via a reward function, including more dexterous tasks for which grasping and reaching may not be sufficient. For example, this made the addition of the cabinet and drawer opening tasks straightforward. We did not have to provide the method with any information about the explicit environment model, object geometries, or demonstration trajectories.
>
>
> **Have these responses adequately addressed your concerns? We value your feedback and look forward to continuing our constructive discussion.**
>
> [1]: “Deep Reinforcement Learning for Robotic Manipulation with Asynchronous Off-Policy Updates”, Gu et al. 2016
>
> [2]: “Demonstration-Guided Reinforcement Learning with Learned Skills”, Pertsch et al. 2021
>
> [3]: “Latent Plans for Task Agnostic Offline Reinforcement Learning”, Rosete et al. 2022

---

> > ### Comment · Reviewer_D5az · 2023-08-13
> > **Read the authors feedback - keeping the same score**
> >
> > I read the authors response. As far as I see the assumptions that I took are according to what I assumed while reviewing the paper. As I already indicated, I find the topic important but the differentiation from previous works and the innovation should be larger.

---

> > > ### Author Response · Authors · 2023-08-15
> > > **Follow-up to D5az**
> > >
> > > Thank you for your timely review of our comment.  We would like to clarify that our work is novel, and differs from prior work in several ways. Our novel contributions include:
> > >
> > > - To our knowledge, the first method to enable a robotic agent to design specialized tools based on the task at hand while simultaneously learning control policies for using them. Prior work that performs joint optimization only optimizes for a single objective such as forward motion [1, 4, 5, 6], or optimizes designs without also considering control [3, 7, 8].
> > > - Eight new simulated robotic tool manipulation tasks for this problem setting, which will help to guide future progress in this direction. Existing works on jointly optimizing design and control tend to focus on locomotion tasks [1, 4, 5, 6, 7].
> > > - Empirical evaluations of the generalization and few-shot finetuning performance of learned designer and controller policies.
> > > - An adjustable tradeoff parameter to enable practical tradeoffs between material and power consumption, whereas prior works focus solely on task success.
> > > - Hardware experiments demonstrating the effectiveness of the designed tools in the real world, as opposed to prior works that evaluate primarily in simulation [1, 2, 4, 5, 6, 8].
> > >
> > >
> > > We hope that this may help address your concerns. We would be happy to continue to discuss any specific prior works or concerns that you may have in mind.
> > >
> > > ---
> > > [1]: "Data-Efficient Co-Adaptation of Morphology and Behavior with Deep Reinforcement Learning", Luck et al. 2019
> > >
> > > [2]: "Imagine That! Leveraging Emergent Affordances for 3D Tool Synthesis", Wu et al. 2019
> > >
> > > [3]: "Fit2Form: 3D Generative Model for Robot Gripper Form Design," Ha et al. 2020
> > >
> > > [4]: "Task-Agnostic Morphology Evolution", Hejna et al. 2021
> > >
> > > [5]: "Embodied Intelligence via Learning and Evolution", Gupta et al. 2021
> > >
> > > [6]: "Transform2Act: Learning a Transform-and-Control Policy for Efficient Agent Design", Yuan et al. 2022
> > >
> > > [7]: "Learning Tool Morphology for Contact-Rich Manipulation Tasks with Differentiable Simulation", Li et al. 2022
> > >
> > > [8]: "Physical Design Using Differentiable Learned Simulators", Allen et al. 2022

---

### Author Response · Authors · 2023-08-16
**Note to AC regarding D5az's Review**

Dear AC,
We would like to make a brief note regarding Reviewer D5az’s most recent comment. Particularly, they mentioned that "As I already indicated, I find the topic important but the differentiation from previous works and the innovation should be larger," but we were not able to find a discussion of this in the initial review. We have requested further clarification from the reviewer, but unfortunately have not yet heard back as the discussion period draws to a close. As a result, we regrettably may not have had the opportunity to address their specific concerns.
Thank you for your time,
Authors

---

### Decision · Program_Chairs · 2023-08-30

**Decision:**

Accept (Poster)

**Comment:**

The paper proposes to design the tool for each specific manipulation task by having separate designer and control policies where the designer policy designs the tool and the control policy performs the task using the tool. The approach is evaluated in both simulation and with a real Franka Panda robot arm. The experiments show sample efficiency and transfer to new tasks.